# Bazedoxifene does not share estrogens effects on IgG sialylation

**Priti Gupta[1,2], Karin Horkeby[2], Hans Carlsten[1], Petra Henning[2], Cecilia Engdahl[1,2]***

**1** Department of Rheumatology and Inflammation Research, Institute of Medicine, Sahlgrenska Academy at University of Gothenburg, Gothenburg, Sweden, **2** Department of Internal Medicine and Clinical Nutrition, Institute of Medicine, Sahlgrenska Osteoporosis Centre, Sahlgrenska Academy at University of Gothenburg, Gothenburg, Sweden

* cecilia.engdahl@gu.se

**Data Availability Statement:** The data supporting this study's findings are openly available in the figshare repository DOI 10.6084/m9.figshare.22313056.

## Abstract

The incidence of rheumatoid arthritis (RA) increases at the same time as menopause when estrogen level decreases. Estrogen treatment is known to reduce the IgG pathogenicity by increasing the sialylation grade on the terminal glycan chain of the Fc domain, inhibiting the binding ability to the Fc gamma receptor. Therefore, treatment with estrogen may be beneficial in pre-RA patients who have autoantibodies and are prone to get an autoimmune disease. However, estrogen treatment is associated with negative side effects, therefore selective estrogen receptor modulators (SERMs) have been developed that have estrogenic protective effects with minimal side effects. In the present study, we investigated the impact of the SERM bazedoxifene on IgG sialylation as well as on total serum protein sialylation. C57BL6 mice were ovariectomized to simulate postmenopausal status, followed by ovalbumin immunization, and then treated with estrogen (estradiol), bazedoxifene, or vehicle. We found that estrogen treatment enhanced IgG levels and had a limited effect on IgG sialylation. Treatment with bazedoxifene increased the sialic acids in plasma cells in a similar manner to E2 but did not reach statistical significance. However, we did not detect any alteration in IgG-sialylation with bazedoxifene treatment. Neither estrogen nor bazedoxifene showed any significant alteration in serum protein sialylation but had a minor effect on mRNA expression of glycosyltransferase in the bone marrow, gonadal fat, and liver.

## 1. Introduction

Rheumatoid arthritis (RA) predominately affects women with a peak incidence at 50–55 years of age, coinciding with the average age of menopause, suggesting a relationship between estrogen (E2) deprivation and the onset of RA [1]. Previously, we have shown that estrogen treatment increases the sialylation of IgG which reduces the pathogenicity of IgG, which is of paramount importance for the induction of RA [2].

Glycosylation is a post-translational enzymatic modification that occurs in the Golgi apparatus and endoplasmic reticulum. Glycosylation influences the stability and biological functions of proteins. Several studies have demonstrated that glycosylation is altered by age [3,4] and is also associated with gender [5].

**Funding:** Swedish Research Council supported salary (CE) and expensive (2019-01852), the Swedish state under the agreement between the Swedish government and the country councils, the ALF agreement supported salary (CE) and expensive (770351 & 965726), Konung Gustav V Foundation supported salary (PG) and expensive, the Swedish Association Against Rheumatism supported salary (PG), OE Edla Johansson Foundation supported expensive and Åke Wiberg Foundation supported salary (CE) and expensive. The funders had no role in study design, data collection, and analysis, decision to publish, or preparation of the manuscript.

**Competing interests:** The authors have declared that no competing interests exist.

Estrogen treatment has several beneficial effects in postmenopausal women including a protective effect against postmenopausal osteoporosis. However, estrogen treatment is also associated with negative side effects such as an increased risk of breast cancer and venous thrombosis [6]. Therefore, selective estrogen receptor modulators (SERMs) have been developed in the search for molecules that exhibit estrogen-positive effects on the bone but devoid of estrogenic side effects like breast cancer and venous thrombosis. SERMs bind to estrogen receptors and induce either agonistic or antagonistic effects in a tissue-dependent manner. Bazedoxifene (BZA) is a SERM that is currently used in clinical practice to counteract postmenopausal osteoporosis, but still has some negative side effects on reproductive organs and venous thrombosis [7,8]. Previous findings have shown that treatment with estrogen and bazedoxifene suppresses experimental arthritis and prevents arthritis-related bone loss [9,10]. Estrogen as well as bazedoxifene are known to inhibit B-lymphopoiesis, and estrogen has also been shown to stimulate antibody production [11–14].

Autoantibodies that respond to self-antigens have diagnostic, predictive, and prognostic value in RA [15]. Most autoantibodies are IgG and become activated after binding to their antigen. IgGs are glycoproteins, and the Fc portion of IgG contains a single conserved N-linked glycosylation site, Asp297, where attachment of glycans, especially the terminal sialic acid is known to influence binding affinity to Fc gamma receptors (FcγRs) and thereby regulating IgG inflammatory effect [16,17] which are associated with various inflammatory diseases [18–20]. There is evidence suggesting that glycosylation is altered with less sialic acids on autoantibodies just before the onset of RA [21].

We have previously shown that estrogen increases IgG Fc sialylation [2]. This may be one factor why susceptibility to RA changes after menopause when estrogen decreases dramatically. The role of SERMs in antibody pathogenicity and their potential use as a treatment option for menopausal women with impending RA has not been previously investigated. A treatment that would counteract the decrease in sialylation of autoantibodies and thereby make the antibodies less pathogenic might inhibit the progression into the clinical phase of RA.

In the present study, we use ovalbumin (OVA)-induced estrogen-deprived ovariectomized (OVX) mice to mimic an inflammatory condition in post-menopausal females to investigate whether treatment with bazedoxifene affects the IgG glycosylation in the same manner as E2. We found that only estrogen treatment increases the levels of IgGs in OVA-induced estrogen-deprived mice. The intensity of sialic acids in bone marrow plasma cells was upregulated by estrogen and a strong tendency with bazedoxifene. However, bazedoxifene did not share estrogens response on sialylation of IgG.

## 2. Materials and methods

### 2.1 Animal care

Female C57BL/6J mice (Taconic) were kept under standard conditions and fed a phytoestrogen-free pellet diet (Harlan) *ad libitum* at the Experimental Biomedicine Gothenburg, Sweden. The study was approved by the ethics committees of the Gothenburg region, Sweden (case number 1–2017). Sample size analysis was performed using the G*power software using data from a previous study for the sialic acid effect on IgG in the OVX-vehicle and OVX-E2 mice groups [2]. With an alpha value of 0.05, a power of 80%, and a calculated Cohen's d effect size of 1.66, our study would require a sample size of 7 per group to achieve significance. The experimental timeline is displayed in S1 Fig.

To stop the production of natural sex hormones and mimic postmenopausal conditions, mice were subjected to ovariectomy surgery at 9 weeks of age under isoflurane inhalation

(Baxter Medical AB) and Metacam (Boehringer Ingelheim Animal Health) as preoperative analgesia.

To induce IgG production, all mice were immunized subcutaneously (*s.c.*) at the tail base on day 0 with 100 μg OVA (Sigma-Aldrich) emulsified in complete Freund's adjuvant (Sigma-Aldrich), followed by a booster injection of 100 μg OVA, emulsified in incomplete Freund's adjuvant (Sigma-Aldrich) on day 28.

Ten days after the OVA immunization, hormone treatment was initiated. Based on the sample size calculation we started the experiment with 10 mice per group. Due to health concerns, a few mice were taken away during the experiment, and this resulted in n = 8/veh, n = 8/E2, and n = 8/BZA. Mice were *s.c.* injected with 17β-estradiol-3-benzoate (E2; 1 μg/mouse/day; Sigma-Aldrich), Bazedoxifene (BZA; 24 μg/mouse/day; Pfizer) dissolved in 100 μl of Miglyol oil or vehicle (100 μl/mice/day; Miglyol oil) 5 times per week for a total of 4 weeks. Treatment doses for E2 and BZA were based on previous findings from our group, which include both healthy ovariectomized C57Bl6 mice [11,22] and in pathogenic conditions such as collagen-induced arthritis a model of RA [10], and in experimental lupus [23]. In addition, the body surface is calculated to ensure the dose of BZA used in mice is comparable to doses used in the relevant to human [24].

Mice were terminated 38 days after the first OVA immunization and after 20 days of treatment, using Ketamine (Richter Pharma) and Dexmedetomidine (Orion Pharma Animal Health) for sedation, blood was collected and euthanized by cervical dislocation. The uterus, thymus, gonadal fat as well as spleen were dissected and weighed.

## 2.2 Flow cytometry (FC) analyses

Single-cell suspensions from bone marrow (BM) were processed in PBS for FC analyses. Cells were counted using a cell counter (Sysmex Europe GmBH). FC staining was performed using fluorochrome-conjugated antibodies, i.e, APC anti-CD267 (TACI), Per-CPanti-CD19, BV421anti-CD138, V500anti-B220 and APC-Cy7anti-CD3, all purchased from eBiosciences. For intracellular staining, cells were fixed with Perm buffer (Thermo Fisher Scientific) and then stained, according to the manufacturer's protocol. Fluorescein isothiocyanate (FITC) conjugated- sambuca nigra lectin (SNA) (Vector Lab), was used to stain intracellular sialic acid. The gating strategy for B-cells, plasma cells, and the SNA+ population are shown in (S3 Fig). Analysis was performed using the BD FACS-verse Flow Cytometer and Flow Jo software (FLowJo10.6.2).

## 2.3 Peripheral quantitative computed tomography (pQCT) analysis

Mice tibia bone was fixed in 4% paraformaldehyde and cortical bone mineral density (BMD) was assessed with a scan in the mid-diaphyseal region by pQCT analysis using Stratec pQCT XCT Research M software (software version 5.4) as previously described [25].

## 2.4 Serum analyses

Total serum was obtained from blood by centrifugation of a tube containing serum gel (Sarsted). Total serum IgG was measured using a commercially available kit (Bethyl Laboratories). OVA-specific IgG was measured using an in-house ELISA as described previously in [2]. Sialic acids on IgG were measured using two different in-house ELISAs. *I)* Lectin ELISA [2]: Briefly, wells coated with anti-mouse IgG-F(ab)2 followed by serum incubation, biotinylated SNA, then streptavidin-HRP, and later reaction development. *II)* Reverse lectin ELISA [26]: Briefly, wells coated with unconjugated SNA (Vector lab), followed by serum incubation, then anti-mouse IgG-HRP, and later reaction development.

Glycolipids and glycoproteins from serum were separated using the detergent extraction method [27,28]. The total sialic acids were measured in total serum, serum glycolipids, and serum glycoproteins using the Sialic acid assay kit (Sigma Aldrich).

## 2.5 RNA isolation and Real-time PCR for mRNA expression

Total RNA was extracted from snap-frozen BM, liver, and gonadal fat tissue using RNeasy Mini Kit (Qiagen). RNA (0.1 μg/sample) was reverse transcribed into cDNA using a cDNA Reverse Transcription synthesis kit (Thermo Fisher). cDNA corresponding to 2.5–5 ng RNA was used for quantitative PCR with TaqMan (Thermo Fisher) and analysis for mRNA expression was performed with the Applied Biosystem Step OnePlus Real-Time PCR using *ST6Gal1* (TaqMan Mm00486119_m1, NCBI Gene ID- 20440), *B4Galt2* (TaqMan Mm00480752_m1, NCBI Gene ID- 53418), and *Fut8* (TaqMan Mm00489795_m1, NCBI Gene ID- 53618) primers and probe set (Thermo Fisher) following TaqMan standard protocol. The housekeeping gene 18S (4310893E, Applied Biosystems) was used as endogenous control, and data were analysed using the delta-delta-CT method and presented as % of the vehicle.

## 2.6 Statistical analysis

Statistical analyses were performed using GraphPad Prism software (version 0.1.0 (216)). We found one significant outlier using Grubb's test from one of the animals in BM for *St6gal1*, *B4Galt2*, and *Fut8* enzymes, which were excluded from the analyses. Groups were compared with a One-way analysis of variance (ANOVA) followed by Dunnett's multiple comparisons against the vehicle group. Student t-test was performed to compare the two groups. Data are presented as mean ± SEM with a scattered bar graph and $p < 0.05$ was considered statistically significant and p values $< 0.10$ was considered a trend.

## 3. Results

### 3.1. Bazedoxifene has an impact on organ weight and bone parameter in ovariectomized OVA immunized mice

To evaluate the effect of E2 and BZA in ovariectomized OVA immunized mice, the weight of the uterus, thymus, gonadal fat, and spleen were measured on termination. We observed that mice treated with E2 have significantly larger uteri while the effect was absent in the BZA group compared to the vehicle (Table 1). On the other hand, E2 and BZA treatment significantly reduced thymus (Table 1) and gonadal fat weight (Table 1). However, E2 increased the

**Table 1. Organ weights and tibia cortical density.**

|  | Uterus (mg) | Thymus (mg) | Gonadal fat (mg) | Spleen (mg) | Tibia Ct. BMD (mg/cm$^3$) |
|---|---|---|---|---|---|
| **Vehicle** | 11.6 ± 1.62 | 70.9 ± 4.98 | 335.8 ± 30.94 | 151.7 ± 8.85 | 1017.2 ± 7.23 |
| **E2** | 192.4 ± 9.96 *** | 15.9 ± 0.92 *** | 133.9 ± 9.44 *** | 182,6 ± 11.44 | 1062.9 ± 4.52 *** |
| **BZA** | 9.5 ± 1.30 | 57.9 ± 2.36 * | 210.7 ± 9.50 *** | 135.6 ± 4.98 | 1047.7 ± 4.47 ** |

Data expressed as mean ± SEM. Significant differences calculated towards vehicle group (One-way ANOVA), =

*p $<$0.05

**p $<$0.01

***p $<$0.00. n = 8 mice/group.

spleen weight (p = 0.06) in a limited manner but this was not shown in BZA treated group (Table 1). To determine the effect of E2 and BZA on bone, we performed pQCT analysis of the tibia bone. As expected, we found significantly higher cortical bone density after treatment with E2 and BZA (Table 1).

### 3.2. Bazedoxifene has no impact on IgG- sialylation or total serum protein sialylation of ovariectomized OVA immunized mice

To access the effect of E2 and BZA on IgG pathogenicity, we investigated several aspects of IgG in the serum. As expected, after 4 weeks of E2 treatment there was a significant induction in total IgG levels compared to vehicle (Fig 1A). This induction was not displayed with BZA (Fig 1A). Further, neither of the treatment altered the OVA-specific IgG levels compared to vehicle (Fig 1B). The sialic acids on all IgGs were investigated using two different ELISAs, E2 treatment showed only a tendency of increased sialic acid levels on IgG but did not reach statistical significance after considering all groups (Fig 1C and 1D). When considering only the vehicle and E2 treated group, E2 treatment significantly induced the sialic acid on IgG (S2A and S2B Fig). No difference was observed in sialic acids of IgG after treatment with BZA compared to the vehicle (Fig 1C and 1D). The sialic acid on anti-OVA-specific IgG was investigated but with no detectable alteration in either of the treatments (S2C Fig). Furthermore, no alteration in sialylation was observed either in total serum or separated glycoproteins or glycolipids by any of the treatments (Fig 1E).

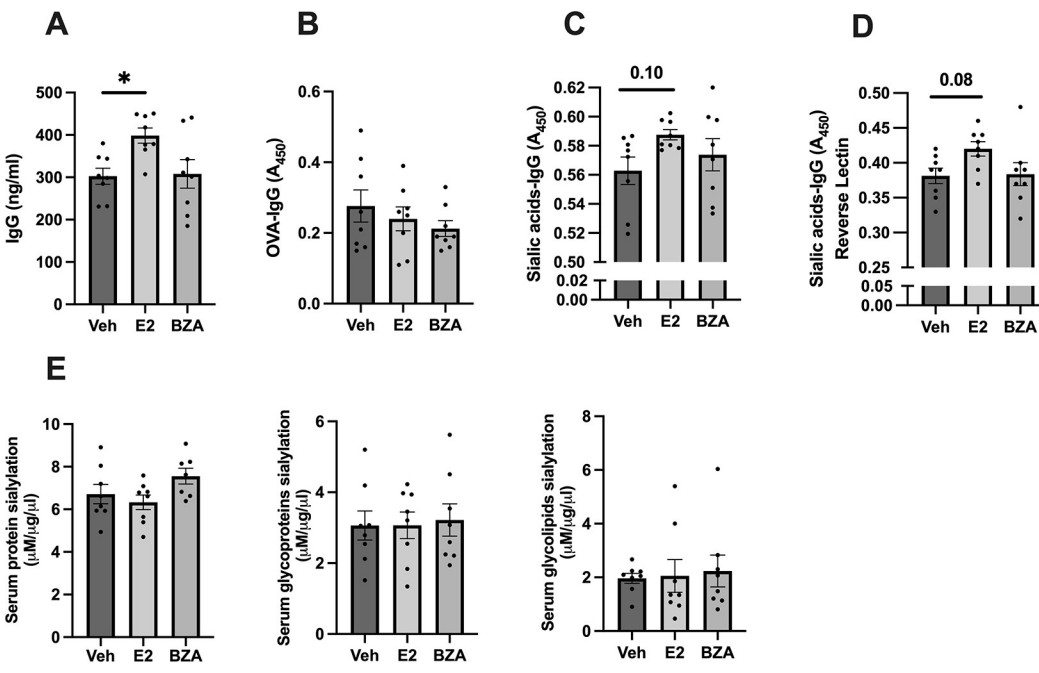

**Fig 1. Impact on the degree of IgG sialylation and total protein sialylation.** (**A**) Total IgG concentration, (**B**) Optical density of OVA-specific IgG, (**C**) Concentration of sialic acid on total IgG (lectin ELISA), (**D**) Concentration of sialic acid on total IgG (reverse lectin ELISA), and (**E**) Concentration of serum protein sialylation, serum glycoproteins sialylation, and serum glycolipids sialylation. One-way ANOVA followed by Dunnett's multiple comparisons to assess differences towards vehicle. $*P < 0.05$. A (450): Absorbance (450 nm). (n = 8/group). E2: 17β-estradiol-3-benzoate, BZA: Bazedoxifene, veh: Vehicle.

### 3.3. Estrogen but not bazedoxifene has an impact on intracellular sialic acid levels in ovariectomized OVA immunized mice

Next, we investigated the E2 and BZA impact at the protein level by checking the levels of intracellular sialic acid levels in BM. We found that BZA treatment of ovariectomized OVA immunized mice did not affect the frequency of B cells in our experimental settings (Fig 2A). However, as expected, E2 significantly reduced the frequency of B cells in BM (Fig 2A). We could not find any alteration in BM-derived plasma cells frequency by E2 or BZA (Fig 2A). None of the treatments altered the sialic acids in all live BM cells (Fig 2B). E2 treatment showed a tendency to increase sialic acid levels in BM-derived B-cells. Intriguingly, E2 significantly increased the intensity of sialic acid levels whereas BZA treatment showed a strong tendency in BM-derived plasma cells (Fig 2B).

### 3.4. Treatment with estrogen and bazedoxifene have minor effects on the mRNA levels of glycosyltransferases in ovariectomized OVA-immunized mice

To determine the impact of E2 and BZA on protein glycosylation enzymes at the mRNA level, we analysed the expression of glycosyltransferases. A slight decrease of *B4Galt2* mRNA in bone marrow was observed in the E2-treated group but not with BZA, and this was not shown for either *St6gal1* or *Fut8* (Fig 3A). We also found that *Fut8* mRNA expression in gonadal fat was significantly reduced with in the E2 treated group, but not that of *St6gal1* or *B4Galt2* (Fig 3B). Whereas, in the liver, we did not detect any change in the mRNA expression of *St6gal1*, *B4Galt2*, or *Fut8* (Fig 3C).

## 4. Discussion

It is well established that autoantibodies precede the symptomatic induction of active RA, and the sialylation status of autoantibodies plays an important role in regulating the pathogenicity [29]. Previously, we have demonstrated that estrogen mediates a direct effect on IgG pathogenicity by upregulating IgG sialylation in both inflammatory-induced mice and RA postmenopausal patients [2]. Implying that estrogen deprivation during menopause leads to a decrease in IgG sialylation, which may promote the induction of active autoimmune diseases. Here, we hypothesized that not only estradiol (E2) but also the SERM bazedoxifene (BZA) affect the IgG

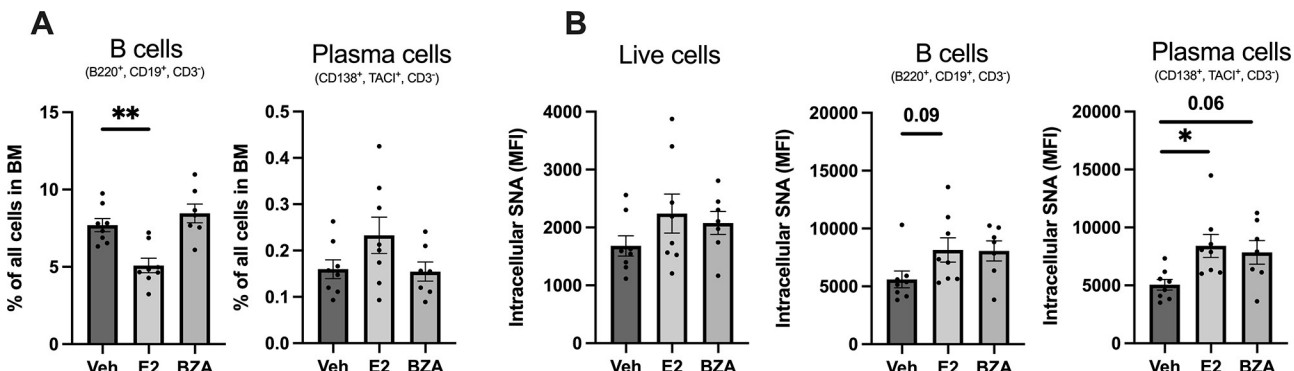

**Fig 2. Impact on the cell frequency and intracellular sialic acid levels.** (**A**) Frequency of bone marrow-derived B- and plasma cells. (**B**) Sialic acid levels in BM-derived total cells, B-cells, and plasma cells were measured by SNA staining. MFI: Mean fluorescence intensity. One-way ANOVA followed by Dunnett's multiple comparisons to assess differences towards vehicle. *$P < 0.05$, (n = 8/group). E2: 17β-estradiol-3-benzoate, BZA: Bazedoxifene, veh: Vehicle.

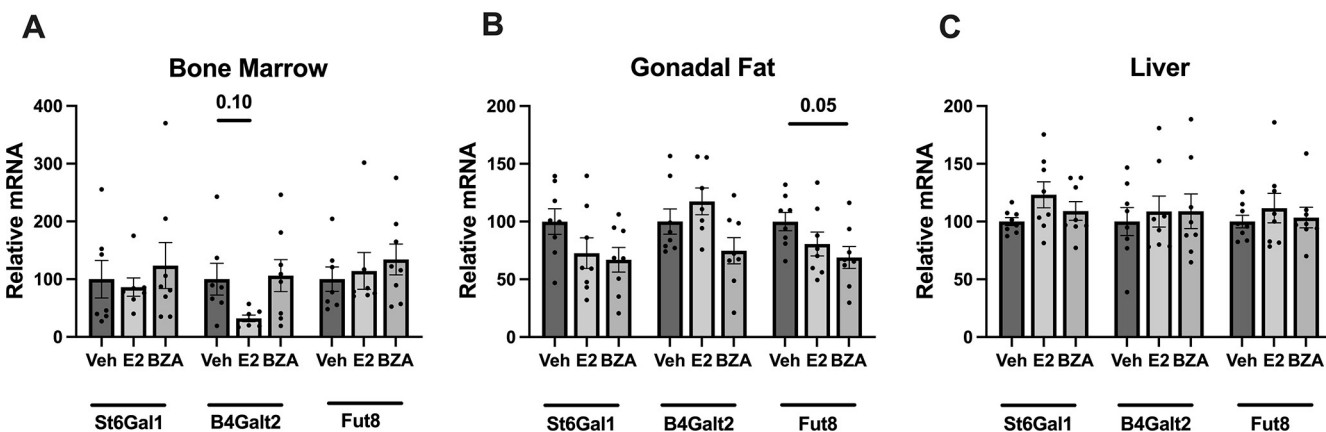

**Fig 3. Effects on relative mRNA expression of glycosyltransferase, α2,6-sialyltransferase1 (*St6Gal1*), β-1,4-galactosyltransferase2 (*B4Galt2*), and α1,6-fucosyl transferase (*Fut8*).** Gene expression in (**A**) bone marrow, (**B**) gonadal fat, and (**C**) liver. One-way ANOVA followed by Dunnett's multiple comparisons to assess the difference towards a vehicle for each analysed gene. (n = 8/group). E2: 17β-estradiol-3-benzoate, BZA: Bazedoxifene, veh: Vehicle.

sialylation. If so, bazedoxifene may be a better treatment option for pre-RA women having autoantibodies at the onset of menopause. In contrast to our hypothesis, here we show that bazedoxifene has no effect on IgG sialylation whereas E2 shows a limited alteration. Interestingly, E2 treatment was shown to increase sialic acid levels in plasma cells whereas bazedoxifene showed a strong tendency. Additionally, neither of the treatment caused any alteration in total protein sialylation.

In the present study of OVA-induced ovariectomized mice, we show that the bazedoxifene does not change uterine weight, as previously reported compared to estrogen treatment [9,10,14]. However, bazedoxifene displays an estrogen agonistic effect by reducing thymic and gonadal fat weight. In line with previous studies, both in immune-induced and naïve ovariectomized mice [9,10,23], we demonstrated that estrogen and bazedoxifene have a stimulatory effect on tibial cortical bone compared with vehicle-treated.

Previous studies have reported that estrogen decreases B cell frequency and increases plasma cell frequency [12,14,30,31]. The same effect on decreased B cell frequency is reported of bazedoxifene in naïve mice [14,30,31] and only the bazedoxifene effect on Ig-producing B cells has previously been investigated [11]. In the present study, we could only confirm the reduction in B-cells by estrogen compared to vehicle, but in contrast to previous findings, we did not observe any alteration of bazedoxifene on B-cell lymphopoiesis. One possible explanation for this might be that the limited effect of bazedoxifene has previously been reported in naïve conditions and is not strong enough to affect the B cells when they are, as in the present study, dramatically induced with OVA immunization. Neither estrogen nor bazedoxifene affects the plasma cell frequency in the OVA-induced mice and just as above the treatment efficiency might not be sufficient to promote the effect.

In this study, estrogen treatment increased serum IgG levels consistent with previously reported data [2,12,32]. However, bazedoxifene responses on serum IgG have not been investigated previously. Here we show no impact of bazedoxifene on IgG response. Additionally, neither estrogen nor bazedoxifene has any effect on OVA-specific IgG antibody levels which confirms the previous study of estrogen effect on OVA-specific IgG [2]. Previously, we have reported that estrogen affects IgG sialylation in inflammatory conditions in both RA patients and OVA-induced mice [2]. Additionally, *Ercan et al.* and *Mijakovac et al.* have shown that estrogen treatment increases IgG glycosylation [33,34] and treatment with phytoestrogen increases IgG glycosylation in experimental arthritis [35]. In this study, we demonstrate that

estrogen treatment displays a tendency to increase IgG sialylation compared to vehicle using two-different ELISAs, but this was not altered by bazedoxifene treatment. When only considering estrogen and vehicle group a significant elevation (t-test p = 0.03) was observed in estrogen treated group. In addition, in this study, neither estrogen nor bazedoxifene, alter the sialylation of OVA specific IgG. Due to certain limitations and three fundamental differences in the experimental design, we did not fully verify the estrogen's effect on IgG sialylation observed by *Engdahl et al 2018* in this study [2]: *I)* The interval between the first and second OVA immunization was, due to ethical regulations, longer (20 days compared to 38 days), which could restrict the immune response. *II)* The treatment duration was shorter (20 compared to 38 days), and treatment was initiated after initial OVA immunization compared to 38 days before in the previous study, which could limit the effect on sialylation by estrogen treatment. *III)* The administration method was different. In *Engdahl et al 2018*, mice were treated with E2 slow-release pellets while the mice received E2 injections in the present study. A previous report from *Ström et al.* suggests that E2 pellets release higher levels during the first weeks which thereafter substantially decreases [36]. Therefore, it is possible that the treatment dose was initially higher in *Engdahl et al 2018*, resulting in a more pronounced effect on the initial IgG production.

Interestingly, we show a significant increase of sialic acid levels in plasma cells by estrogen and a strong tendency with bazedoxifene treatment. The less impact of bazedoxifene in the level of sialic acid in plasma cells might be the treatment dose which may not be enough to induce the sialic acid level significantly.

Previously, it has been reported that sialylation of serum glycoproteins is increased in postmenopausal women and continues to increase with age after menopause [3,5]. This implies that estrogen influences glycosylation. Consistent with previous studies [3,5], we could not find any alteration of sialic acid in total serum protein by any of the treatments, suggesting that sialic acid levels in serum protein are independent of estrogen status. A recent study in experimental arthritis shows that phytoestrogen treatment increases the mRNA expression of sialyltransferases in splenic tissue [35]. Additionally, we and others have shown that estradiol treatment stimulates glycosyltransferases in plasmablasts and B cells *in vitro* [2,34]. In contrast, *Nelson et al* showed that estrogen does not affect the glycosyltransferases in the liver [37]. In accordance with *Nelson et al*, we did not observe any effect of estrogen and bazedoxifene, on the mRNA levels of sialyltransferases in the liver and the same pattern was also seen in BM and gonadal fat suggesting that the general protein sialylation is independent on the estrogen status. However, there is a slight correlation between glycosyltransferase expression levels and glycan levels, indicating that regulatory mechanisms may be more composite than a simple change in the glycosyltransferase expression [34,38].

In summary, this is the first study where we have studied the effects of the SERM bazedoxifene on IgG pathogenicity and on total protein glycosylation. In this study, we have shown that bazedoxifene shares some of the estrogenic characteristics including effects on organs and bones. However, we found no effect of bazedoxifene treatment on IgG sialylation and thereby antibody pathogenicity in OVA-induced estrogen-deprived mice mimicking the inflammatory condition observed in postmenopausal RA patients. Furthermore, our finding that bazedoxifene lacks some estrogenic characteristics may be of importance in terms of using it as a treatment option for postmenopausal women with autoantibodies.

## Supporting information

**S1 Fig. Experimental setup for ovalbumin (OVA)-immunization in ovariectomized (OVX) mice.** Female C57Bl6 mice were ovariectomized at 9 weeks of age (day -10). On day 0 mice

were 1st immunized with OVA and complete Freund's adjuvants (CFA) subcutaneously. On day 10, treatment was initiated with hormones. Mice received 2nd immunization with OVA and incomplete Freund's adjuvant (IFA) on day 28. Mice were terminated on day 38. (TIFF)

**S2 Fig. Impact on the degree of IgG sialylation and anti-OVA IgG sialylation. (A)** the concentration of sialic acid on total IgG (lectin ELISA), **(B)** the concentration of sialic acid on total IgG (reverse lectin ELISA) Student's t-test used to assess the difference. (C) Sialic acid on OVA-IgG. One-way ANOVA followed by Dunnett's multiple comparisons to assess the difference towards a vehicle. *$P < 0.05$, A (450): Absorbance (450 nm). (n = 8/group). E2: 17β-estradiol-3-benzoate, BZA: Bazedoxifene, veh: Vehicle. (TIFF)

**S3 Fig. Gating strategy for flow cytometry in bone marrow B cells.** Singlets were determined using FSC-Height versus FSC-Area followed by live cell gating based on size and granularity. CD3− cells were gated into B- cells (double positive of CD19+ and B220+), and plasma cells (CD138+ TACI+). In live, B and plasma cells sialic acids were detected intracellular using mean fluorescence intensity (MFI) of sambuca nigra lectin (SNA). (TIFF)

## Acknowledgments

The authors acknowledge the excellent technical assistance from Dr Jianyao Wu, Dr Tibor Sághy, and Ms Carolina Johansson.

## Author Contributions

**Conceptualization:** Priti Gupta, Hans Carlsten, Petra Henning, Cecilia Engdahl.

**Data curation:** Priti Gupta, Karin Horkeby, Hans Carlsten, Petra Henning, Cecilia Engdahl.

**Formal analysis:** Priti Gupta, Petra Henning, Cecilia Engdahl.

**Funding acquisition:** Hans Carlsten, Cecilia Engdahl.

**Investigation:** Priti Gupta, Petra Henning, Cecilia Engdahl.

**Methodology:** Priti Gupta, Karin Horkeby, Petra Henning, Cecilia Engdahl.

**Project administration:** Hans Carlsten, Petra Henning, Cecilia Engdahl.

**Resources:** Cecilia Engdahl.

**Supervision:** Hans Carlsten, Petra Henning, Cecilia Engdahl.

**Validation:** Priti Gupta, Petra Henning, Cecilia Engdahl.

**Visualization:** Priti Gupta, Karin Horkeby, Petra Henning, Cecilia Engdahl.

**Writing – original draft:** Priti Gupta, Petra Henning, Cecilia Engdahl.

**Writing – review & editing:** Priti Gupta, Karin Horkeby, Hans Carlsten, Petra Henning, Cecilia Engdahl.

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
