## [Decision Letter · Decision Letter 0]

15 Mar 2023

PONE-D-23-02656Bazedoxifene does not share estrogens effects on IgG sialylationPLOS ONE

Dear Dr. Engdahl,

Thank you for submitting your manuscript to PLOS ONE. After careful consideration, we feel that it has merit but does not fully meet PLOS ONE’s publication criteria as it currently stands. Therefore, we invite you to submit a revised version of the manuscript that addresses the points raised during the review process.

We look forward to receiving your revised manuscript.

Kind regards,

Masanori A. Murayama

Academic Editor

PLOS ONE

Journal Requirements:

   "The authors received no specific funding for this work" 

Additional Editor Comments:

Thank you for submitting your interesting study. Authors needs response for many critical comments from editor and reviewers. I have required an additional control, non-ovariectomized mice in all experiments. And why authors did not investigated enzyme activity in B cells ? E2 and BZA really have activity ? It may be inactivated ? The given dose of E2 and BZA is optimization ?

Reviewers' comments:

Reviewer's Responses to Questions

**Comments to the Author**

1. Is the manuscript technically sound, and do the data support the conclusions?

Reviewer #1: Partly

Reviewer #2: Partly

2. Has the statistical analysis been performed appropriately and rigorously? 

Reviewer #1: N/A

Reviewer #2: Yes

3. Have the authors made all data underlying the findings in their manuscript fully available?

Reviewer #1: Yes

Reviewer #2: Yes

4. Is the manuscript presented in an intelligible fashion and written in standard English?

Reviewer #1: Yes

Reviewer #2: Yes

5. Review Comments to the Author

Reviewer #1: The paper entitled “Bazedoxifene does not share estrogens effects on IgG sialylation” investigated effects of bazedoxifene and E2 on the IgG sialylation. However, several issues should be addressed:

1- The initial description of the abstract is not complete and coherent.

2- This study requires a control group. Why didn't you study a control group in addition to the Vehicle group?

3- How was the sample size calculated?

4- How many mice were in each group?

5- In result section, you said "Neither of the treatment affects the mRNA levels of glycosyltransferases in ovariectomized OVA immunized mice" but in figure 3 there is significance difference. Also the text of lines 188-192, does not coordinate with the diagrams A and B (about significant signs of the figure 3).

6- In discussion section this sentence need references: "Consistent with previous studies, we could not find any alteration of sialic acid in total serum protein by any of the treatments, suggesting that sialic acid levels in serum protein are independent of estrogen status."

Reviewer #2: The MS "Bazedoxifene does not share estrogens effects on IgG sialylation" is well written and nicely presented. However, there are few major points the authors need to address before it can be considered for publication

Major:

1. Did the authors measure sialic acid levels on anti-OVA antibodies? If not, this is required to correct for the preexisting sialysation.

2. In addition to measure sialation, did the authors check the animals for RA symptoms? If not, it is important to measure sialation and correlate with RA.

3. Authors performed all experiments at a single dose, and they have discussed this. They need to repeat at least IgG sialation experiments at few higher doses. It is not justified to draw a conclusion without exploring few more doable experimental settings.

Minor:

1. Mention in methods the number of mice/group.

6. PLOS authors have the option to publish the peer review history of their article (what does this mean?). If published, this will include your full peer review and any attached files.

Reviewer #1: No

Reviewer #2: No

---

## [Author Response · Author response to Decision Letter 0]

5 Apr 2023

Editors Comment

Thank you for submitting your interesting study. Authors need responses for many critical comments from editors and reviewers. I have required an additional control, non-ovariectomized mice in all experiments. And why authors did not investigate enzyme activity in B cells? E2 and BZA really have activity? It may be inactivated. The given dose of E2 and BZA is optimization?

1. I have required an additional control, non-ovariectomized mice in all experiments.

Answer: This is a good point requested by the editor. Our interest in this paper was to build on the findings from Engdahl et al. Arthritis and Research Therapy 2018 by demonstrating that not only estrogen treatment but also SERM Bazedoxifene (BZA) could inhibit estrogen-deficient loss of sialic acid on IgG, which affects the binding ability to Fc gamma receptors. We could see that estrogen increases the sialic acid on IgG and thereby reduces the pathogenicity, but we could not show that bazedoxifene has the same estrogenic effect. In the paper written by me, Engdahl et al. Arthritis and Research Therapy 2018, I show that estrogen deficiency, OVX, reduced the IgG-Fc sialic acids compared to non-ovariectomized mice, but comparing the estrogen-deficient induction was the scope of this manuscript.

2. Why authors did not investigate enzyme activity in B cells? E2 and BZA really have activity? It may be inactivated.

Answer: Estrogens regulation capacity on murine plasma blast form splenocytes, as well as human plasma cells from PBMC, was previously demonstrated in the publication by Engdahl et al. Arthritis and Research Therapy 2018. Due to technical issues in determining the appropriate dose of BZA, we were unable to repeat this experiment in the present manuscript. Therefore, we do not know whether BZA shares this estrogenic effect on plasmablasts in Vitro. However, we indeed observe the same pattern of estrogen stimulation in splenocyte-derived plasma blasts. Additionally, we did show that SNA, sialic acids altered only limited by E2 treatment in B cells and significantly increased by E2 and a strong tendency by BZA in plasma cells. Further, none of the treatments had any effect on the levels of ST6Gal1 mRNA expression in either BM, gonadal fat, or liver. 

3. The given dose of E2 and BZA is optimization?

Answer: Thank you for your comment and we really appreciate it. The dose of E2 and BZA is optimized. The used dose of E2 and BZA in this study, has previously been published from our group in ovariectomized healthy C57Bl6 mice, by Bernardi et al. Immunity, Inflammation and Disease 2015, Gustafsson et al. Journal of Endocrinology 2022, and in pathogenic condition collagen-induced arthritis (RA), Andersson et al. Rheumatology 2015, as well as in experimental lupus (SLE) Nordqvist et al. Lupus 2022. To ensure the relevance to humans, we calculated the doses of BZA based on body surface area (Reagan-Shaw et al, 2008 FASEB J). This part is now included in the material and method section of the revised version of the manuscript.

Material and method; lines 99-104; “Treatment doses for E2 and BZA were based on previous findings from our group, which include both healthy ovariectomized C57Bl6 mice (11, 22) and in pathogenic conditions such as collagen-induced arthritis a model of RA (10), and in experimental lupus (23). In addition, the body surface is calculated to ensure the dose of BZA used in mice is comparable to doses used in the relevant to human (24).”

Reviewers’ comments

Reviewer #1: The paper entitled “Bazedoxifene does not share estrogens effects on IgG sialylation” investigated the effects of bazedoxifene and E2 on IgG sialylation. However, several issues should be addressed:

Major comments

1. The initial description of the abstract is not complete and coherent.

Answer: I appreciate the reviewer bringing this up. The abstract has now been improved and became complete and coherent. We have provided both a clean version and the updated manuscript, along with marked changes.

Abstract; lines 15-31; “The incidence of rheumatoid arthritis (RA) increases at the same time as menopause when estrogen level decreases. Estrogen treatment is known to reduce the IgG pathogenicity by increasing the sialylation grade on the terminal glycan chain of the Fc domain, inhibiting the binding ability to the Fc gamma receptor. Therefore, treatment with estrogen may be beneficial in pre-RA patients who have autoantibodies and are prone to get an autoimmune disease. However, estrogen treatment is associated with negative side effects, therefore selective estrogen receptor modulators (SERMs) have been developed that have estrogenic protective effects with minimal side effects. In the present study, we investigated the impact of the SERM bazedoxifene on IgG sialylation as well as on total serum protein sialylation. C57BL6 mice were ovariectomized to simulate postmenopausal status, followed by ovalbumin immunization, and then treated with estrogen (estradiol), bazedoxifene, or vehicle. We found that estrogen treatment enhanced IgG levels and had a limited effect on IgG sialylation. Treatment with bazedoxifene increased the sialic acids in plasma cells in a similar manner to E2 but did not reach statistical significance. However, we did not detect any alteration in IgG-sialylation with bazedoxifene treatment. Neither estrogen nor bazedoxifene showed any significant alteration in serum protein sialylation but had a minor effect on mRNA expression of glycosyltransferase in the bone marrow, gonadal fat, and liver.”

2. This study requires a control group. Why didn't you study a control group in addition to the Vehicle group?

Answer: We are uncertain about which type of control has been requested. However, as addressed in the response to the editor, we were only interested to investigate the effect on the post-menopausal status and therefore did not include a sham-operated control. The difference between estrogen deficiency, OVX, and sham was displayed in Engdahl et al. Arthritis and Research Therapy 2018. Here in the present study, we showed that estrogen deficiency, OVX, induced a lower degree of sialic acids on IgG, a phenotype that increased the IgG pathogenicity.

3. How was the sample size calculated?

Answer: We acknowledge the reviewer's comment. Because this manuscript was submitted to a journal requiring less text, we removed the sample size calculation. We have now included the sample size calculation in the revised manuscript.

Material and method; lines 82-86; “Sample size analysis was performed using the G*power software using data from a previous study for the sialic acid effect on IgG in the OVX-vehicle and OVX-E2 mice groups (2). With an alpha value of 0.05, a power of 80%, and a calculated Cohen’s d effect size of 1.66, our study would require a sample size of 7 per group to achieve significance.”

4. How many mice were in each group?

Answer: We had 8 mice/ group. n=8 /vehicle, n=8 /E2, and n=8 /BZA. Now we have included the number of mice per group in the material and method of the revised version of the manuscript.

Material and method; lines 94-97; “Based on the sample size calculation we started the experiment with 10 mice per group. Due to health concerns, a few mice were taken away during the experiment, and this resulted in n= 8/veh, n=8/E2, and n=8/BZA.”

5. In result section, you said "Neither of the treatment affects the mRNA levels of glycosyltransferases in ovariectomized OVA immunized mice" but in figure 3 there is significance difference. Also the text of lines 188-192, does not coordinate with the diagrams A and B (about significant signs of the figure 3).

Answer: Thank you for bringing this to our attention. We also appreciate the reviewer noticing the variations in glycosyltransferase levels caused by the various treatments. The result outcome statement 3.4 and the description for this result have now been modified in the revised versions of the manuscript.

Results; line 194-203 “ 3.4. Treatment with estrogen and bazedoxifene has minor effects on the mRNA levels of glycosyltransferases in ovariectomized OVA immunized mice: 

To determine the impact of E2 and BZA on protein glycosylation enzymes at the mRNA level, we analysed the expression of glycosyltransferases. A slight decrease of B4Galt2 mRNA in bone marrow was observed in the E2-treated group but not with BZA, and this was not shown for either St6gal1 or Fut8 (Fig. 3A). We also found that Fut8 mRNA expression in gonadal fat was significantly reduced with a p-value of p= 0.05 in the E2 treated group, but not that of St6gal1 or B4Galt2 (Fig. 3B). Whereas, in the liver, we did not detect any change in the mRNA expression of St6gal1, B4Galt2, or Fut8 (Fig. 3C).”

6. In the discussion section this sentence needs references: "Consistent with previous studies, we could not find any alteration of sialic acid in total serum protein by any of the treatments, suggesting that sialic acid levels in serum protein are independent of estrogen status."

Answer: Thank you for bringing this to our attention. The reference to this sentence has now been added to the revised version of the manuscript.

Discussion: Line 266-269 “This implies that estrogen has an effect on glycosylation. Consistent with previous studies ( 3, 5), we could not find any alteration of sialic acid in total serum protein by any of the treatments, suggesting that sialic acid levels in serum protein are independent of estrogen status.” 

Reviewer:2 2

Major comments

1. Did the authors measure sialic acid levels on anti-OVA antibodies? If not, this is required to correct for the preexisting sialyation.

Answer: We really appreciated the reviewer's point, regarding checking the sialic acid on anti-OVA-IgG. We did look at sialic acid on anti-OVA IgG and we did not see any alternation. Now we have included the data as supplementary fig 2C and also discussed the result in the revised version.

Result: line 179-181; “The sialic acid on OVA-specific IgG was also investigated but we did not detect any change in either of the treatments (Supplementary Fig. 2 C).”

Discussion: lines 247-248; “In addition, in this study, neither estrogen nor bazedoxifene, alter the sialylation of OVA-specific IgG.”

2. In addition to measure sialation, did the authors check the animals for RA symptoms? If not, it is important to measure sialation and correlate with RA.

Answer: It would be interesting to look at a RA animal model, but our study was not performed as a RA model, and therefore we did not look for RA symptoms. To understand the pathogenicity of IgG, we needed to massively induce the IgG response, which we did in this experiment with OVA immunization. The alteration of the IgG pathogenicity is indeed important in the development of autoimmune disease and especially in RA. 

3. Authors performed all experiments at a single dose, and they have discussed this. They need to repeat at least IgG sialation experiments at few higher doses. It is not justified to draw a conclusion without exploring few more doable experimental settings.

Answer: Thank you for your comment, which we appreciate. We respectfully disagree that a higher dose of BZA would be clinically relevant, as our dosages were carefully selected based on previous publications. In our group, it has previously been published in ovariectomized healthy C57Bl6 mice, by Bernardi et al. Immunity, Inflammation and Disease 2015, Gustafsson et al. Journal of Endocrinology 2022, and in pathogenic condition collagen-induced arthritis (RA), Andersson et al. Rheumatology 2015, as well as in experimental lupus (SLE) Nordqvist et al. Lupus 2022. To ensure relevance to humans, we calculated the doses of BZA based on body surface area (Reagan-Shaw et al, 2008 FASEB J). This part is now lifted in material and method to the revised version of the manuscript.

Material and method; lines 99-104; “Treatment doses for E2 and BZA were based on previous findings from our group, which include both healthy ovariectomized C57Bl6 mice (11, 22) and in pathogenic conditions such as collagen-induced arthritis a model of RA (10), and in experimental lupus (23). In addition, the body surface is calculated to ensure the dose of BZA used in mice is comparable to doses used in the relevant to human (24).”

While a higher dose of BZA may produce a more pronounced effect, it is also likely to lead to alterations in uterine weight. The purpose of using BZA is to engage estrogen’s positive effect, such as on bone, without the negative effect of inducing weight in reproductive organs. The aim of this present study was to investigate the ability of BZA to reduce the pathogenic capability of IgG, which could be the basis for future clinical studies in pre-RA patients with autoantibodies and limited clinical signs of RA, as well as the post-menopausal patient from hinder them to develop clinical symptoms. However, this follow-up study will not investigate the effect of BZA on IgG glycosylation. 

Minor:

3. Mention in methods the number of mice/group.

Answer: Thank you for bringing this to our attention. We have now added the number of mice in the revised version of the manuscript.

Material and method; lines 94-97; “Based on the sample size calculation we started the experiment with 10 mice per group. Due to health concerns, a few mice were taken away during the experiment, and this resulted in n= 8/veh, n=8/E2, and n=8/BZA.”

---

## [Decision Letter · Decision Letter 1]

2 May 2023

Bazedoxifene does not share estrogens effects on IgG sialylation

PONE-D-23-02656R1

Dear Dr. Cecilia,

We’re pleased to inform you that your manuscript has been judged scientifically suitable for publication and will be formally accepted for publication once it meets all outstanding technical requirements.

Kind regards,

Masanori A. Murayama

Academic Editor

PLOS ONE

Additional Editor Comments (optional):

Thank you for submitting your revision. I will endorse publish your manuscript. However, I suggest that you revise your manuscript according to reviewers comments.

Reviewers' comments:

Reviewer's Responses to Questions

**Comments to the Author**

1. If the authors have adequately addressed your comments raised in a previous round of review and you feel that this manuscript is now acceptable for publication, you may indicate that here to bypass the “Comments to the Author” section, enter your conflict of interest statement in the “Confidential to Editor” section, and submit your "Accept" recommendation.

Reviewer #1: All comments have been addressed

Reviewer #2: All comments have been addressed

2. Is the manuscript technically sound, and do the data support the conclusions?

Reviewer #1: Yes

Reviewer #2: Yes

3. Has the statistical analysis been performed appropriately and rigorously? 

Reviewer #1: N/A

Reviewer #2: Yes

4. Have the authors made all data underlying the findings in their manuscript fully available?

Reviewer #1: Yes

Reviewer #2: Yes

5. Is the manuscript presented in an intelligible fashion and written in standard English?

Reviewer #1: Yes

Reviewer #2: Yes

6. Review Comments to the Author

Reviewer #1: Thanks for the corrections you made.

Regarding the answer to point 5, it seems that you should write the BZA treated group instead of the E2 treated group in this sentence: "We also found that Fut8 mRNA expression in gonadal fat was significantly reduced with a p-value of p= 0.05 in the E2 treated group, but not that of St6gal1 or B4Galt2 (Fig. 3B)."

Reviewer #2: Minor comment

Table 1. E2 data for Spleen is 182,6 � 11.44. If so, it is significantly more than that for vehicle, and should be mentioned with statistical significance.

7. PLOS authors have the option to publish the peer review history of their article (what does this mean?). If published, this will include your full peer review and any attached files.

Reviewer #1: No

Reviewer #2: No

---

## [Editor Report · Acceptance letter]

8 May 2023

PONE-D-23-02656R1 

Bazedoxifene does not share estrogens effects on IgG sialylation 

Dear Dr. Engdahl:

I'm pleased to inform you that your manuscript has been deemed suitable for publication in PLOS ONE. Congratulations! Your manuscript is now with our production department. 

Kind regards, 

on behalf of

Dr. Masanori A. Murayama 

Academic Editor

PLOS ONE